# Biological Interfacial Materials for Organic Light-Emitting Diodes

**DOI:** 10.3390/mi14061171

**Published:** 2023-05-31

**Authors:** Amjad Islam, Syed Hamad Ullah Shah, Zeeshan Haider, Muhammad Imran, Al Amin, Syed Kamran Haider, Ming-De Li

**Affiliations:** 1Department of Chemistry and Key Laboratory for Preparation and Application of Ordered Structure Materials of Guangdong Province, Shantou University, Shantou 515063, China; 2Department of Applied Physics, E-ICT-Culture-Sports Convergence Track, College of Science and Technology, Korea University-Sejong Campus, Sejong City 30019, Republic of Korea; syedhamadullahshah@korea.ac.kr; 3Department of Civil and Environmental Engineering, Yonsei University, Seoul 03722, Republic of Korea; chemistudent@gmail.com; 4Chemistry Department, Faculty of Science, King Khalid University, P.O. Box 9004, Abha 61413, Saudi Arabia; imranchemist@gmail.com; 5Department of Electrical Engineering, College of Engineering, Gyeongsang National University, Jinju-si 52828, Republic of Korea; alamin.pkst@gmail.com; 6Department of Chemistry, Chung-Ang University, 84 Heukseok-ro, Dongjak-gu, Seoul 06974, Republic of Korea; haederseoul@gmail.com

**Keywords:** biological, interfacial materials, organic light-emitting devices

## Abstract

Organic optoelectronic devices have received appreciable attention due to their low cost, mechanical flexibility, band-gap engineering, lightness, and solution processability over a broad area. Specifically, realizing sustainability in organic optoelectronics, especially in solar cells and light-emitting devices, is a crucial milestone in the evolution of green electronics. Recently, the utilization of biological materials has appeared as an efficient means to alter the interfacial properties, and hence improve the performance, lifetime and stability of organic light-emitting diodes (OLEDs). Biological materials can be known as essential renewable bio-resources obtained from plants, animals and microorganisms. The application of biological interfacial materials (BIMs) in OLEDs is still in its early phase compared to the conventional synthetic interfacial materials; however, their fascinating features (such as their eco-friendly nature, biodegradability, easy modification, sustainability, biocompatibility, versatile structures, proton conductivity and rich functional groups) are compelling researchers around the world to construct innovative devices with enhanced efficiency. In this regard, we provide an extensive review of BIMs and their significance in the evolution of next-generation OLED devices. We highlight the electrical and physical properties of different BIMs, and address how such characteristics have been recently exploited to make efficient OLED devices. Biological materials such as ampicillin, deoxyribonucleic acid (DNA), nucleobases (NBs) and lignin derivatives have demonstrated significant potential as hole/electron transport layers as well as hole/electron blocking layers for OLED devices. Biological materials capable of generating a strong interfacial dipole can be considered as a promising prospect for alternative interlayer materials for OLED applications.

## 1. Introduction

One of the most viable solutions for realizing the increasing global energy requirements with sustainable and renewable resources involves the incorporation of natural materials for the fabrication of energy conversion/storage devices. In this respect, extensive attempts are being made on the development of new organic light-emitting diodes (OLEDs) [1,2]. These devices possess many advantages of organic compounds, which can be easily modified to obtain the desirable device performance [3,4,5,6]. After decades of research, organic semiconductors have started to show superior electronic properties. This has resulted in the construction of novel organic devices with amazing performances and high efficiency [7,8,9,10,11,12,13,14,15]. Moreover, the low-cost organic materials processed through facile solution processing methods are very intriguing and fascinating for commercial applications of lightweight, flexible and transparent OLED devices [16].

The performance of OLED devices relies principally on the nature and properties of the active layer and electrodes, but also depends on the properties of the device interfaces to a great extent. In the past few years, extraordinary progress has been achieved in device performance through efficient interfacial engineering, i.e., by inserting well-defined interfacial materials between the active layers and electrodes [17,18]. Among various types of interfacial materials, organic and solution-processable materials are of particular interest for academic and industrial scientists. Generally, the interfacial layer performs three common roles in OLED devices [19]. Firstly, it facilitates reducing the energy level mismatch at the interface by increasing or reducing the work function (WF) of the electrode (anode/cathode). In principle, a low WF metal cathode is desirable in a conventional device structure, whereas a high WF anode metal is desirable in an inverted structure device. Nevertheless, high WF metals (Pt, Au) are highly expensive, whereas low WF metals (Ba, Ca) are very reactive and toxic. Hence, to mitigate these issues, and to utilize the more feasible metal electrode (Al or Ag), an interfacial layer is needed to alter their WF.

Secondly, the defects and traps present at the surface of the active layer can be mitigated by using an interfacial layer between the electrode and active layer. These defects are often produced from either chemical interaction or thermal damage and/or structural imperfection, which results in low device stability [20,21,22]. An organic interfacial layer introduces some charged/polar groups (zwitterions) to eliminate the defects/traps at the interface. 

Thirdly, interfacial layers can significantly enhance the functional properties of film (such as conductivity and wettability). A rough surface of thin film can hamper the charge transportation and the introduction of interfacial layers will not only reduce the surface roughness but also enable better wettability through minimizing the surface tension [19].

Conclusively, interfacial materials prepared from simple and facile methods can facilitate in boosting the overall device performance, and also exhibit good compatibility with many fabrication procedures [23]. Most importantly, adding multiple layers in OLED devices increases not only the complexity but also the overall cost of fabrication. Thus, considerable effort is being focused primarily on the discovery of high-performance, low-cost interfacial materials [24,25]. Over the last few years, biological interfacial materials have been widely investigated to decrease the dependence on petroleum-based materials and decrease environmental pollution [26,27].

Among the several bio-sourced materials, biological interfacial materials (BIMs)— generally obtained from animals, microorganisms and plants—have appeared as impressive alternatives owing to their biocompatibility, biodegradability, cost-effectiveness, sustainability and eco-friendly nature [28,29]. Several examples of BIMs are ampicillin, deoxy-ribonucleic acid (DNA), nucleobases and sulfonated-lignin (Figure 1). Outstanding features such as versatile structures, rich functional groups (-OH, -COOH, and -NH_2_) and generation of an interfacial dipole make biological materials effective as efficient interfacial layers for OLEDs [30,31,32].

As far as the utilization of biological materials in energy and environmental applications is concerned [33,34,35], their electronic potential in energy conversion/storage devices remains unexplored and substantial research is needed to understand completely the mechanism and charge dynamics of BIMs and how they improve the performance of energy conversion/storage devices. Additionally, biological materials generally exhibit very low solubility and conductivity [36], and it is a highly difficult task to functionalize and modify them so they can serve as interfacial layers for electronic devices [37,38]. However, their natural abundance, biocompatibility, low-cost, eco-friendly nature and effectiveness emphasize their immense potential for the construction of efficient organic energy conversion/storage devices.

In this paper, we present a detailed review of the various biological materials employed in OLEDs. First, we briefly introduce different biological materials and how their intrinsic properties make them beneficial for OLED devices. We then address the basic working mechanism of an OLED device, and how it is changed and improved by incorporating biological materials in the device architecture. Lastly, we present a comprehensive review of the application of biological materials in OLEDs. We expect that this review will be an important contribution to summarize the key methods of minimizing environmental pollution and realizing sustainable development. Likewise, it will assist in streamlining future developments in the nascent area of “green” optoelectronics.

## 2. Biological Materials

Recently, several research articles have been published on the applications of bio-based organic energy conversion/storage devices. Nevertheless, in the last few years, important progress has been made in the performance of energy conversion/storage devices by employing green biological materials to pursue sustainability issues [39,40,41]. In this segment, we highlight a brief depiction of biological materials that have been applied as interfacial layer materials in OLED devices only (Figure 1).

### 2.1. Ampicillin

Ampicillin is a renowned biomaterial exploited for animal healthcare. Ampicillin belongs to a class of beta-lactam antibiotics known as penicillin, which is employed to cure several diseases such as bladder infections, gonorrhea, meningitis, pneumonia and other stomach/intestines infections. Ampicillin possesses in vitro biological activity against the Gram-positive and Gram-negative aerobic/anaerobic bacteria. The chemical structure of ampicillin consists of three types of amines (primary, secondary and tertiary) and a carbonyl (-C=O) group. Ampicillin generates a large interfacial dipole (30.28 D) due to the presence of polar functional moieties (amines and carbonyl) which assign a spatial charge distribution in its molecule.

### 2.2. Lignin and Lignosulfonates

Lignin and lignosulfonates (LS) are water-soluble biopolymers prepared as by-products in the process of sulfite pulping [42]. The total production of lignin throughout the world is almost 60–70 million tons per year [39]. The major portion of the lignin and lignosulfonates is obtained from paper and is utilized to generate electricity [43]. Generally, the most frequent application of lignin is the preparation of thermosetting materials, and it is also used in phenolic resins for wood-bonding [44]. Lignin’s aromatic structure (Figure 1) and electron-transfer characteristic have recently outlined attention for its possible application in organic electronics, particularly in PVSCs, OSCs and OLEDs [31,43]. Many aromatic rings in lignin and its derivatives absorb well in the ultra-violet region of electromagnetic spectrum. The oxidation of electron-rich molecules is related to hole transportation ability in organic electronic devices. In LS, its phenol oxidation and the J-aggregation characteristic provide it with superior hole transport properties [44].

### 2.3. Deoxyribonucleic Acid (DNA)

DNA is also a biopolymer that includes the critical genetic instructions for the conformation of proteins and ribonucleic acid (RNA) in the shape of a double helix architecture (Figure 1) [45]. The monomers of DNA are known as nucleotides; each nucleotide is composed of a pentose sugar, a phosphate group and a nitrogenous base. The backbone of the nucleotide is negatively charged (phosphate), which is surrounded by sodium (Na^+^) and hydrogen (H^+^) cations [45]. This polarity of DNA molecule enables it to be dissolved in the polar solvents and helps its solution processability. The DNA used for electronic devices is largely received from natural sources (salmon fish) [46]. The conventional DNA extraction method involves multiple steps that yield a relatively small quantity of DNA. Moreover, DNA exhibits high thermal stability up to 1400 °C in its solid state [47,48]. The fascinating properties of DNA, such as suitable energy levels (lowest-unoccupied-molecular-orbital (LUMO): −1.1 eV; highest-occupied-molecular-orbital (HOMO): −5.2 eV), makes it feasible to be employed in organic electronic devices as a hole-transport layer (HTL) and an electron-blocking layer (EBL) [46,49,50,51]. 

### 2.4. Nucleobases (NBs)

Nucleobases (NBs) of DNA are also known as nucleic acid bases (such as adenine (A), cytosine (C), guanine (G) thymine (T) and uracil (U)). RNA is a single-stranded nucleic acid biopolymer that serves to covert the DNA base sequence into different proteins. RNA consists of A, C, G, and U bases. The NBs can be obtained from renewable materials and can also be prepared synthetically, positioning them as a much economical alternative to DNA and other conventional organic optoelectronic materials. These are basically small molecules that can be employed for device fabrication through the vacuum vapor deposition method without further purification/modification. Additionally, NBs possess a simple molecular structure and much lower molecular weight compared to DNA polymers, which makes their processing very easy.

## 3. Working Mechanism of OLEDs and Significance of Biological Interfacial Materials

### 3.1. Working Mechanism of OLEDs

The basic and simple architecture of an OLED is composed of organic electron transport layer (ETL) and HTLs sandwiched between two electrodes. The interface between the two layers obtained a decisive role in achieving balanced charge recombination and, consequently, produced pure and stable electroluminescence (EL). Disregarding the interfacial layers, Figure 2a displays the general bi-layer configuration of a standard OLED device. By applying a potential difference between the electrodes (anode and cathode), holes are inserted from the anode into the HOMO level of the HTL, whereas the electrons are inserted from the cathode into the LUMO level of the ETL. These charges (holes/electrons) are combined either at the interface of the HTL or ETL, leading to the generation of excitons. The excitons can decay either non-radiatively (by losing energy in the form of heat) or radiatively (by emitting light). 

The basic configuration of OLEDs can also be converted to a tri-layer structure by the addition of an emissive layer (EML) between the HTL and ETL [52]. The prime objective of this layer is to provide a site for the recombination of a hole and electron and hence produce EL. In this respect, the function of each organic layer is well-defined and can be amplified separately. This also reveals that EMLs can be chosen to achieve a desired luminance efficiency as well as the color of EMLs. The extremely thin organic EML facilitates in realizing decreased resistance and provides higher current densities for an applied electrical bias [53].

Over the past two decades, OLEDs have developed into one of the most consistently investigated research areas in the field of material science/engineering [54]. Major uses of OLED devices involve low-resolution display technologies (cellphones, wristwatches, biosensors) as well as high resolution large-area technologies (flat-panel displays). The maximum performance of an OLED device is generally realized by adapting a multi-layered configuration (including HTL, EBL, EML and ETL). Even though EMLs and electrodes are the most important parts in any OLED device structure, interfacial engineering has also obtained a crucial value in realizing high efficiency and device stability [55]. One of the several key causes of low device efficiency is the presence of large charge injection barrier due to the energy-level mismatch between the electrodes and EMLs [56,57,58]. Specifically, the charge injection from an inorganic layer/material into an organic layer/material is an intimidating task. A large injection barrier results in a high turn-on voltage (*V_ON_*), consequently generating an imbalanced transportation of charges, which decreases the device efficiency [59,60,61]. Hence, this creates a serious hurdle in the progress of efficient, stable and economical OLEDs. An impressive way to modify interfacial properties in OLED devices includes the application of a thin interfacial material between EMLs and the electrodes. Therefore, various interfacial materials (PFN, ZnO, MoO_3_, zwitterions) have been exploited to enhance the efficiency of OLEDs [54].

### 3.2. Significance of Biological Interfacial Materials

As of present, as far as the use of biological materials in OLED devices is concerned, it mainly includes ampicillin, DNA, nucleobases (NBs) and lignin compounds (Figure 3) [32,46,50,62,63,64]. Several years ago, our group pioneered ampicillin as a dopant for the conventional PEDOT:PSS HTL layer [65]. Ampicillin is a widely used biomaterial/antibiotic for animal healthcare [66]. The asymmetrical structure of ampicillin enables it to generate an interfacial dipole. The presence of amino groups in ampicillin, LS and NBs is responsible for their interaction with PEDOT:PSS via hydrogen bonding [65]. The interaction of biological precursors with PEDOT:PSS as well as active layer are illustrated in Figure 2b,c.

DNA is another essential BIM for OLEDs. It is a poly-anionic biomaterial with a phosphate group in the side chain through which it interacts with cationic electron-transport/light-emitting materials. Thus, DNA can accomplish an advantageous role as a BIM to improve the performance of OLEDs [51]. The motivation behind the role of DNA as a BIM originates from its very low HOMO level (0.9 eV), supplying a strong energy barrier for the transportation of electrons from cathode at the ETL/EML interface. DNA is mostly employed with a surfactant (CTMA) in the form of a complex (DNA–CTMA). The mode of action of the DNA–CTMA complex is rather different than that of free DNA. Even though the DNA–CTMA complex can serve as both a HTL and ETL, it is more suitable to be employed as a HTL compared to an ETL due to its shallow LUMO energy level [50]. Moreover, the electrical conductivity is obtained due to DNA strands only in the DNA–CTMA complex, whereas the CTMA moiety does not play any role in the transportation of charges [67].

In principle, the water solubility of the DNA–CTMA complex is very poor [32,67], and the utilization of CTMA also makes the fabrication process complicated. In light of these reasons, NBs BIMs were proposed for OLED devices. NBs are simple-structured small molecules that have a very lower molecular weight compared to DNA and possess similar HOMO and LUMO levels (3.6 to 4.1 eV) to that of DNA. Additionally, they can also be easily applied through the vapor deposition process. Nevertheless, their electron-affinity is relatively different than DNA; NBs possessing low ionization potentials act well as an electron blocking layer, whereas those exhibiting high ionization potentials serve well as a HBL [32].

The application of lignin BIM in OLEDs is inspired by the existence of its multiple hydroxyl phenol groups that impart excellent hole-transport characteristics. A crucial lignin-compound is the alkyl chain sulfobutylated lignosulfonate (ASLS), which shows low acidity compared to PEDOT:PSS and possesses distinctive hole transport properties. To make use of these beneficial properties of lignin derivatives, Li et al. synthesized a water-dispersed HTL (PEDOT:ASLS) by substituting PSS with ASLS in pure a PEDOT:PSS layer [64]. The strong oxidation and aggregation nature of ASLS contributed significantly to enhancing the hole injection ability of PEDOT:ASLS, which resulted in a high current efficiency (CE) of 37.65 cd A^−1^ of an OLED device using PEDOT:ASLS as the hole injection layer (HIL).

## 4. State-of-the-Art and Recent Progress in BIM-Based OLED Devices

### 4.1. Ampicillin

In 2019, our group integrated ampicillin (Figure 3) into PEDOT:PSS, not only to upgrade the performance of OLED devices (Table 1), but also to extend the range of green electronics [65]. The amino groups in ampicillin make it stable in acidic conditions and facilitate to generate a large interfacial dipole owing to the asymmetrical conformation of ampicillin. We exploited different ampicillin concentrations and observed that the WF of PEDOT:PSS is reduced by increasing the ampicillin concentration into PEDOT:PSS (Figure 4). The addition of 25% ampicillin into PEDOT:PSS not only established an efficient J-aggregation, but also generated a strong interfacial dipole and improved hole injection in device. The type of aggregation was changed by varying the ampicillin concentration (Figure 4a). Interestingly, the device efficiency was raised to a large extent exhibiting green light with the CE of 120 cd A^−1^, a power efficiency (PE) of 70 lm W^−1^ and an external quantum efficiency (EQE) of 35% [65]. This enhanced efficiency of device based on ampicillin-modified PEDOT:PSS was ascribed to the formation of a horizontal interfacial dipole and J-aggregated excitons (Figure 4a–d). A higher ampicillin concentration (>25%) decreased the device efficiency because of ampicillin crystallization [65].

Inspired by these results, the same group have very recently optimized the annealing conditions to reduce the variations in the efficiency and stability of ampicillin-based top-emission OLEDs (TEOLEDs) [68]. By changing the annealing conditions, the formation of Amp-microstructure (Amp-MSs) of different sizes and shapes (*α*-/*β*-phase) led to different energy states. Amp-MSs excited the light-scattering/out-coupling of the device, which resulted in the reduction of the waveguide modes [68]. The TEOLED device exhibited a record-high EQE (maximum: 68.7% and average: 63.4% at spectroradiometer; maximum: 44.8% and average: 42.6% at integrating sphere) with a wider color gamut (118%) due to the red-shifted J-aggregated emission. The exceptional increase in efficiency was assigned to the improved charge balance and out-coupling, formation of an interfacial dipole, photoluminescence through radiative energy transfer (RET) and EL by J-/H-aggregated excitons (Figure 4e,f) [68]. 

Apart from this, Ali et al. also recently employed ampicillin in InP/ZnSe_x_S_1–x_/ZnS-based quantum-dot light-emitting diodes (QLEDs) as an HTL and ETL at the same time [72]. They achieved a balanced charge distribution through band-alignment, which assisted the generation of an interfacial dipole. The authors maintained the polarity of ampicillin by changing the pH of the charge transportation layers. In ZnO-based ETLs, ampicillin showed anionic behavior at nearly pH 7.5 and created a weak interfacial dipole which reduced the conductivity of ETL. Ampicillin exhibited a cationic nature at almost 4 pH in a PEDOT:PSS-based HTL and created a strong interfacial dipole which boosted the conductivity of the HTL. The QLED device with the configuration of (ITO/(0–75%)ampicillin–PEDOT:PSS/poly[(9,9-dioctylfluorenyl-2,7-diyl)-co-(4,4-(N-(4-sec-butylphenyl)diphenylamine)](TFB)/red-InP-QDs/(0–0.5%)ampicillin:ZnO-NPs/Al) demonstrated almost five times enhancement in the EQE (from 0.90% to 4.70%) due to a balanced charge recombination via energy band alignment [72].

### 4.2. DNA

A thin film of DNA was used initially as a host material for emitters suggesting its possible application in Bio-LEDs. Nevertheless, it could not improve the device performance to an appreciable extent compared to other conventional OLEDs [73,74]. Followed by that, charge carrier properties of DNA were further examined employing multiple device structures. For example, Steckl et al. employed a DNA-based EBL for use in OLED devices [50]. They received the DNA from salmon fish DNA [75], and further processed it to synthesize a DNA–CTMA complex [75]. A Bio-LED emitting green light was constructed using the following layers: ITO/PEDOT:PSS/DNA–CTMA/N,N-bisnaphthalene-1-yl-N,N-bisphenylbenzidine(NPB)/Alq_3_/2,9-dimethyl-4,7-diphenyl-1,10-phenanthrolin (BCP)/Alq_3_/LiF/Al. In both OLEDs (green and blue), an enhanced EL efficiency was obtained using a DNA–CTMA complex as an EBL. The fabricated Bio-LEDs exhibited luminous efficiencies of 8.2 and 0.8 cd A^−1^ (Figure 5a and Table 1) [50].

After a couple of years, Dai et al. fabricated an efficient multi-layered white polymer light-emitting diode (WPLED) using DNA–CTMA as an HTL/EBL. The WPLED was constructed with a device architecture PEDOT:PSS/poly-TPD/DNA–CTMA/MEH-PPV/Cs_2_CO_3_/LiF/Al [69], showing a low *V_ON_* of ~ 5 V, CE of approximately 10.0 cd A^−1^ and brightness of approximately 10500 cd m^−2^ with improved color stability. After showing superior performance in mono-color and white OLED devices, the DNA–CTMA complex was employed as an HTL/EBL to fabricate efficient quantum dot light-emitting devices (QDLEDs). The DNA–CTMA-based QDLEDs were constructed with the configuration of ITO/PEDOT:PSS/poly-TPD/DNA–CTMA/QDs/TPBi/Alq_3_/Ca/Al and realized a low *V_ON_* of 2.6 cd A^−1^, brightness of 6580 cd m^−2^ and a CE of 4.0 cd A^−1^ [70]. This efficient QD-LED demonstrated an improved charge transport with excellent color purity. Later on, DNA was also integrated with polyaniline to form a metal complex (DNA/Polyaniline/Ru(bpy)_3_^2+^) to be employed as an HTL for an OLED device [76]. An OLED with a configuration ITO/DNA/Polyaniline/Ru(bpy)_3_^2+^/Alq_3_/Al showed green light with a high *V_ON_* of 5 V. By accelerating voltage, the emission was altered from green to yellow (14 V) to orange (16 V) and finally to red (18 V). An appreciable role of green emission was due to Alq_3_ and an important role of red emission originated from Ru(bpy)_3_^2+^. 

The basic principle of DNA-based EBL is its very low LUMO energy-level (0.9 eV) [75], and so it gives a considerably high energy barrier for electrons at its interface with an EML. A similar approach was followed by Madhwal et al. [67] who constructed an orange polymer LED (PLED) using MEH-PPV and a blue PLED using PFO with a DNA–CTMA interfacial layer [67]. An orange PLED with an ITO/PEDOT:PSS/DNA–CTMA/MEHPPV/LiF/Al structure and a blue PLED with the configuration ITO/PEDOT:PSS/DNA–CTMA/PFO/LiF/Al were fabricated with an optimized thickness of an EBL interfacial layer (20 nm). They achieved a thickness sensitive improvement in emission intensity of PLED devices with DNA–CTMA BIMs. At high thickness, scattered orientations of DNA molecules were observed, and various pinholes were found at low film thickness. The orange PLED showed improvement in luminance from 30 cd m^−2^ to 100 cd m^−2^, whereas a corresponding blue PLED exhibited enhancement from 80.0 cd m^−2^ to 160.0 cd m^−2^ without/with an EBL interfacial layer, as reported in [67]. After one year, Bazan et al. discovered the application feasibility of DNA as a cathode interfacial layer for PLED devices [51]. The devices were made with the configuration ITO/PEDOT:PSS/MEHPPV/DNA–CTMA/LiF/Al and steered to a substantial rise in efficiency from 0.0065 cd A^−1^ to 0.15 cd A^−1^ (Figure 5b). The authors proposed the generation of an interfacial dipole layer that effectively modifies the WF of an Al cathode [51]. 

### 4.3. Nucleobases (NBs)

To broaden the class of green electronic materials, two NBs-based BIMs (thymine and adenine) (Figure 3) were explored as interfacial layers to raise the efficiency of OLED device [62]. Before exploring the potential of NBs as BIMs in OLED, they were also introduced into OFETs [77,78,79]. Thymine and adenine bases are among the four nitrogenous bases that constitute a large DNA polymer. Thymine belongs to the pyrimidine class and contains one heterocyclic ring, whereas adenine is a member of purine class and possesses two fused rings. These NBs do not involve surfactant treatment or wet processing to make thin films. Green phosphorescent OLED using thymine as an EBL/HTL obtained 76.0 cd A^−1^ CE, which demonstrated a 200% improvement in efficiency compared to the device without NB (37.0 cd A^−1^) (Figure 5c,d) [62]. whereas the adenine-based device also surpassed the reference device with a maximum CE of 48.0 cd A^−1^. A low efficiency roll-off was also observed at higher voltages, resulting in enhanced efficiency (Figure 5c,d) and a similar roughness was found in both bases (1.76 and 1.83 nm) [62]. An AFM scan of adenine showed a more uniform distribution of crystallites with respect to periodicity and height-distribution. Thymine crystallites exhibit columnar-like structures, whereas adenine crystallites possess a lower height dispersion. A better performance of the thymine-based device is ascribed to a coalition of energy levels and structured surface morphology which results in efficient hole transportation to the EML. Inspired by these findings, the same group studied several more NBs (adenine, guanine, cytosine, thymine and uracil) (Figure 3) as EBL and HBL materials for phosphorescence OLEDs (Figure 6a,d) [32]. The low-temperature evaporation process of small NBs generates excellent thin films, providing a smooth integration into the fabrication of OLEDs. NBs make an impact on OLED efficiency on the basis of their electron affinity trend; guanine < adenine < cytosine < thymine < uracil. Guanine and adenine have low electron-affinities (1.8–2.2 eV) than guanine and adenine, avoiding transportation of electrons but accelerating transportation of holes. However, cytosine, thymine and uracil, having high electron-affinities (of 2.6–3.0 eV), can transport electrons while avoiding transportation of holes. By employing adenine as a biological material in an OLED device, a current efficiency (CE) of 52.0 cd A−1 and an external quantum efficiency (EQE) of 14.30% were achieved. This represents an approximate 50% improvement in efficiency compared to the device utilizing an EBL, as seen in [32].

One year later, Steckl et al. made another effort to exploit natural compounds for the preparation of economical, biodegradable and sustainable OLED devices [63]. To create a flexible and conductive anode, researchers introduced a gold film electrode on a cellulose-based substrate. Additionally, in the OLED structure, adenine was employed as a HIL. The gold film showed good adhesion properties and serves as a smooth layer for a flexible substrate, resulting in a uniform light in an OLED device. The versatile behavior of DNA and NBs BIMs has positioned them as being advantageous for green electronics. Flexible OLEDs using an adenine BIM were built using the following architecture: substrate (glass or cellulose)/Au (20 nm)/adenine (10 nm)/NPB (17 nm)/CBP:Ir(ppy)_3_ (30 nm,10 wt%)/BCP (12 nm)/Alq_3_ (25 nm)/LiF (<1 nm)/Al (40 nm). The results revealed that adenine BIMs can decrease the problems associated with the charge injection into the organic EML. Maximum luminance and efficiency of OLED devices on both substrates (glass; from ∼12,500 to 45,000 cd m^−2^ and from 5 to ∼32 cd A^−1^ and cellulose; from ∼2000 to 8400 cd m^−2^ and from 3 to ∼14 cd A^−1^) was magnified owing to the improved hole injection [63].

### 4.4. Sulphonated Lignin (SL)

Interested by the excellent hole transportation capability of the PEDOT dispersed with sulfobutylated lignin (ASLS) (Figure 3) for OSC [80], Li et al. developed a water-soluble ASLS compound that possesses flexible and alkyl-sulfonic groups, exhibiting lower acidic content compared to PEDOT:PSS [64]. PEDOT:ASLS HIL with good water solubility was formed using ASLS as a dopant. For the construction of a blue PhOLED, the PEDOT:ASLS HIL was deposited using a solution process technique. An OLED with the structure of ITO/HIL/TAPC (20 nm)/mCP (8 cm)/mCP:FIrpic (10 wt%, 25 nm)/TmPyPB (35 nm)/LiF (1 nm)/Al (100 nm) was fabricated and realized a CE and PE of 37.65 cd A^−1^ and 12.84 lm W^−1^, as seen in (Figure 6e,f) [64]. These findings suggested that ASLS possesses a promising potential to be employed as a BIM in OLED devices. Significantly, PSS has a non-conjugated structure, whereas ASLS possesses a conjugated structure, and it can be exploited as an HTL. Subsequently, the same group utilized alkali lignin from pulping black liquor to synthesize a grafted sulfonated-acetone-formaldehyde lignin (GSL) (Figure 3) via graft sulfonation [71]. An exceptional cluster-induced green-fluorescence of a GSL polymer was observed. The application potential of GSL as an HTL was examined and GSL exhibited a hole mobility of 2.27 × 10^−6^ cm^2^ V^−1^ s^−1^ [71]. Utilizing the hole-transport ability of GSL, PEDOT:GSLs were prepared and used as HTLs in PhOLEDs with the structure glass/ITO/HTL/TAPC/MCP/MCP:Flrpic/TmPyPB/LiF/Al. The device achieved a maximum PE of 14.67 lm W^−1^ with PEDOT:GSL-1:6, which was 1.78 times higher compared to the device with PEDOT:PSS (8.25 lm W^−1^) (Figure 6g–i) [71]. An improved hole injection, low acidic content, and excellent conductivity of PEDOT:GSL films has resulted in an enhanced OLED device performance.

Conclusively, the presence of hole/electron transport layers (HTL/ETLs) is crucial to prepare efficient OLEDs. Biological materials possess several functional moieties and can be employed to boost the efficiency of an OLED device. Among all the BIMs studied in OLEDs till date, DNA has been examined more often than others as HTL/ETLs and sustained excellent results with a minimum 100% efficiency enhancement. More or less similar results have been realized using NBs BIMs. However, very few NBs have been investigated as BIMs in OLEDs. Furthermore, lignin and its various compounds have been exploited as a HTL in devices and provided nearly 50% increase in device performances with a better stability than the reference devices. A couple of years ago, ampicillin was applied as a dopant for PEDOT:PSS and established a useful BIM for an OLED device with 100% increase in EQE.

## 5. Advantages of BIMs

To date, various types of interfacial materials have been explored for application in OLED devices. These interfacial materials involve metals (Ag, Au, Ca, Al, LiF) and metal oxides (TiO_2_, ZnO, CsCO_3_) possessing low WF. Even though these interfacial materials enhanced the efficiency of OLED devices, they face some serious issues. For instance, metals (Ag, Au, LiF) are prone to air and moisture [54]. However, these are inorganic interfacial materials by nature and their energy levels do not match well with the energy levels of organic layers, which lead to poor charge injection. Moreover, these are expensive materials and require complicated methods for their processing. Compared to inorganic materials, organic materials are receiving excessive attention as interfacial layers because of multiple advantages (easy solution processability, facile preparation, inexpensive, tunable energy-levels and good stability) [54]. Recently, BIMs have shown excellent results and significantly boosted device efficiency [32,46,50,62,63,64]. Some OLED devices containing BIMs (DNA and lignin) have exhibited better stability than that of conventional interfacial materials-based devices [50,62,63,64]. Biological materials possess more structural modification probability compared to other synthetic organic interlayer materials. Additionally, they are abundant, eco-friendly, biodegradable, sustainable and inexpensive (Figure 7). However, a very small number of biological interfacial materials have been applied in OLEDs as compared to other interfacial materials (inorganic and organic). Despite the growing progress of BIMs (synthesis and device application), much more extensive studies and investigations are needed to evaluate their complete mechanism of WF modification of electrodes by BIMs. 

## 6. Conclusions and Outlook

The use of bio-sourced and non-toxic materials is necessary in OLED devices for the evolution of green electronics. We have studied the latest developments in the applications of biological materials as interfacial layers in OLEDs. Biological materials have developed as an advanced class of electronic materials to be exploited as interfacial layers, owing to their excessive availability in nature, low-cost, non-toxicity, and biocompatibility. A very few biological materials have been introduced as interfacial layers in OLEDs as compared to photovoltaic (OSCs and PVSCs) devices. Among different biological materials used ass interfacial layers in OLED devices, ampicillin, DNA, lignin (and its derivatives) and NBs are applicable [32,51,62,64,65,70,71].

Although DNA has the ability to serve both as HTL and ETLs, it has proved better as HTL and EBLs owing to its shallow LUMO level. DNA has also been utilized as a complex (DNA–CTMA) in conjunction with a surfactant called CTMA, which improves its solubility. In addition, NBs also boost the device performance and facilitate the hole transportation owing to their deep HOMO energy levels. The OLED devices using DNA and NBs BIMs demonstrated almost a 100 % rise in device performance. Moreover, sulphonated-lignin compounds (SL, ASL, GSL) have also demonstrated good potential as a substitute of PSS moiety in the pristine PEDOT:PSS and can be exploited as dopants for PEDOT. Lignin-based BIMs boosted the efficiency of OLEDs up to 50% with better stability than reference devices. Recently, the integration of ampicillin as a dopant for HTLs in OLEDs has tremendously enhanced the device efficiency. This suggests the ability of various biological materials to be employed as interfacial layers in OLED applications.

An enhanced performance stability of OLEDs can be realized by overcoming the challenge of device degradation evoked by environmental, operational and fabrication-process parameters [81]. Firstly, environmental degradation of devices generally takes place due to the exposure of devices to oxygen and moisture. Encapsulation with a suitable sealant has been employed to solve this problem. Secondly, operational degradation is also an important problem caused by the WF mismatch between the electrodes and connected organic layers, which leads to the production of pinholes, and heat is produced internally through triplet–triplet annihilation (TTA) or triplet–polaron annihilation (TPA) [81]. Lastly, from the processability point of view, several factors (substrate cleaning, solvent and annealing process) also play a key role to increase the stability of OLEDs. BIM-based devices have exhibited excellent stability compared to the reference (conventional device) [82]. However, more work needs to be completed to investigate their long-term stability potential for commercial applications.

Although promising progress has been made in the development of BIMs, further extensive studies/investigations are needed to gain a complete understanding of charge dynamics and WF modification mechanism of electrodes by BIMs. More aspects still need to be studied to fully understand the role of BIMs in device performance. In general, biological materials do not show good conductivity nor solubility. To make them functional as interfacial layers for OLED devices is a crucial task. The stability of BIM-based devices needs to be improved further and more biological materials that have high stability should be applied as BIMs in electronic devices. The findings regarding the use of BIMs in OLED applications highlight the potential for efficient charge transport properties and the opportunity for the development of green/natural electronics through the easy and cost-effective fabrication of devices using biological materials.

## Figures and Tables

**Figure 1 micromachines-14-01171-f001:**
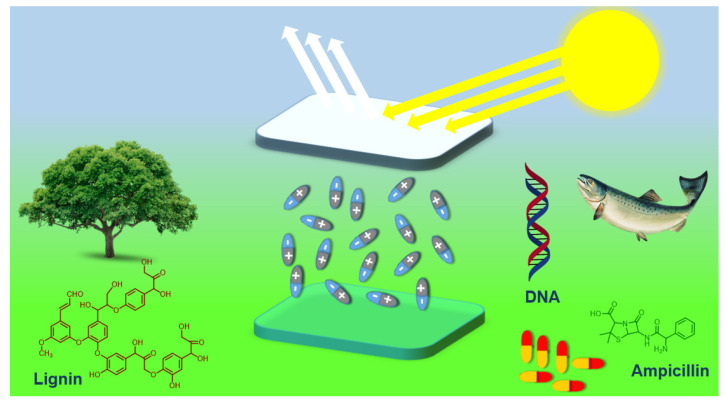
General structures of biological materials (and their source) which are used as interfacial layers for organic light-emitting diodes.

**Figure 2 micromachines-14-01171-f002:**
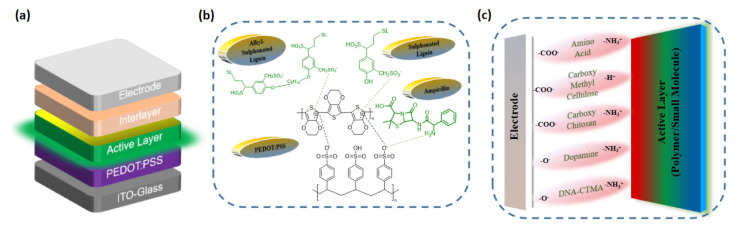
Device structure of (**a**) OLED. Interaction of BIMs with (**b**) PEDOT:PSS; (**c**) active layer in OLED devices.

**Figure 3 micromachines-14-01171-f003:**
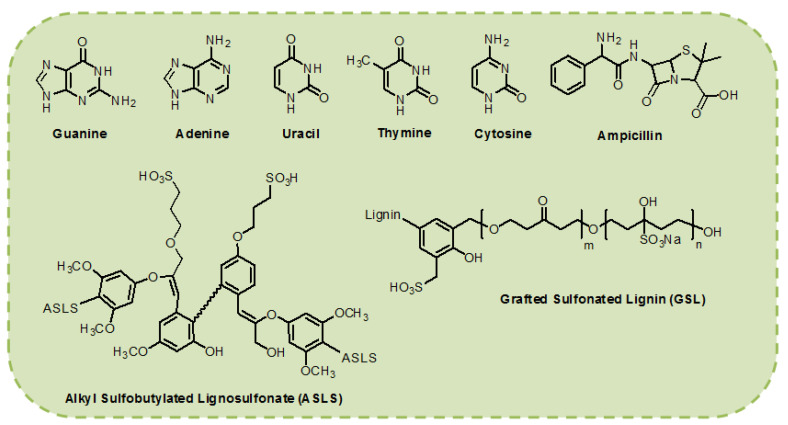
Chemical structures of biological materials used as interfacial layers for the development of OLED devices.

**Figure 4 micromachines-14-01171-f004:**
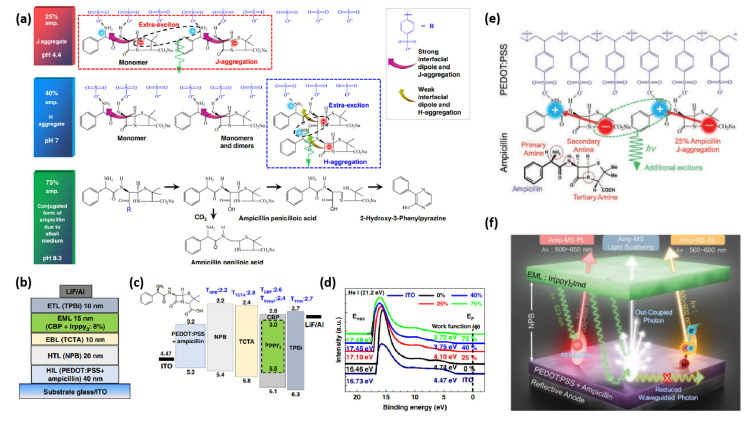
Electrical and optical properties of ampicillin-PEDOT:PSS layers (0–75%). (**a**) Schematic depiction of organic light-emitting diodes (OLEDs) using ampicillin (0–75%) in PEDOT:PSS. (**b**) Band-gap alignment of the constituent layers along with their triplet energies in the device configuration. (**c**) Ultra-violet photoelectron spectroscopy (UPS) analysis of indium tin oxide (ITO)/ampicillin-PEDOT:PSS layers. (**d**) Schematic depiction of the chemical interaction of ampicillin (0–75%) with PEDOT:PSS. Reproduced with permission [65]. Copyright 2019 Springer Nature. (**e**) Chemical interaction of Ampicillin with PEDOT:PSS via hydrogen bonding. (**f**) Schematic illustration of efficiency improvement mechanism of Amp-TEOLEDs through multiple photon-harvesting. Reproduced with permission [68]. Copyright 2022 Wiley.

**Figure 5 micromachines-14-01171-f005:**
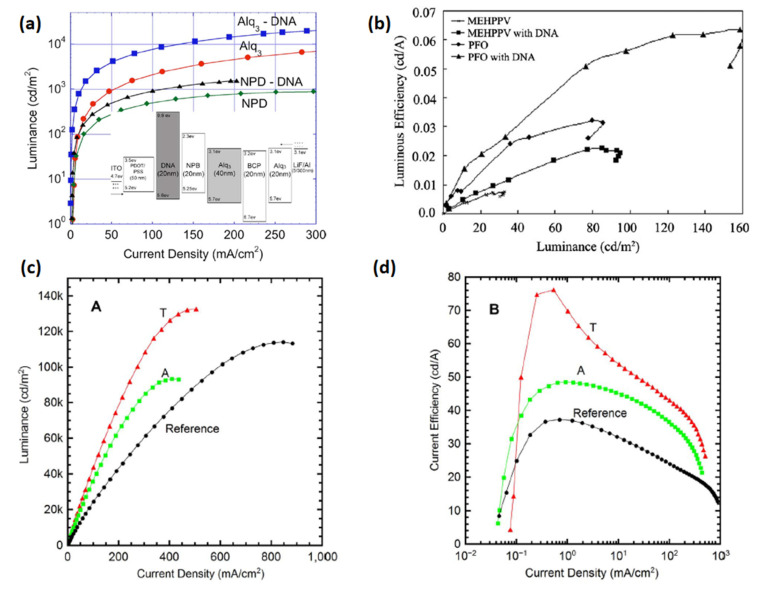
(**a**) Luminous efficiency vs. current density characteristics in EL devices containing BIMs. Reproduced with permission [50]. Copyright 2006 American Institute of Physics. (**b**) Luminous efficiency vs. luminance characteristic of MEH-PPV/PFO-based OLED devices using BIMs. Reproduced with permission [67]. Copyright 2010 Elsevier B.V. (**c**). Luminance vs. current density characteristic of OLEDs employing NBs BIM. (**d**) Current efficiency vs. current density characteristic of NBs-based OLEDs. Reproduced with permission [62]. Copyright 2014 Springer Nature.

**Figure 6 micromachines-14-01171-f006:**
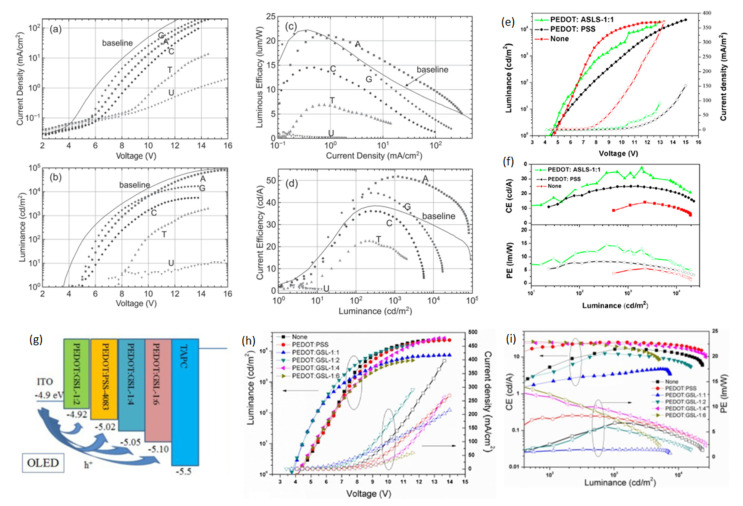
Device characteristics (NB EBL/HTL) (**a**–**d**): (**a**) Current density vs. voltage; (**b**) Luminance vs. voltage; (**c**) Luminous efficiency vs. current density; (**d**) Current efficiency vs. luminance. Reproduced with permission [32]. Copyright 2015 Wiley-VCH. (**e**) Current density vs. voltage plot (hollow symbols) and luminescence vs. voltage plot (solid symbols) of PEDOT:ASLS-based device. (**f**) Current/power efficiency vs. luminance plots of OLEDs with of PEDOT:ASLS-based device. Reproduced with permission [64]. Copyright 2016 American Chemical Society. (**g**) Device structure of the phosphorescent OLEDs. (**h**) The J–V plot and brightness–voltage plot of OLEDs with PEDOT:PSS and PEDOT:GSLs. (**i**) The current and power efficiency plots of OLEDs with PEDOT:PSS and PEDOT: GSLs. Reproduced with permission [71]. Copyright 2016 Royal Society of Chemistry.

**Figure 7 micromachines-14-01171-f007:**
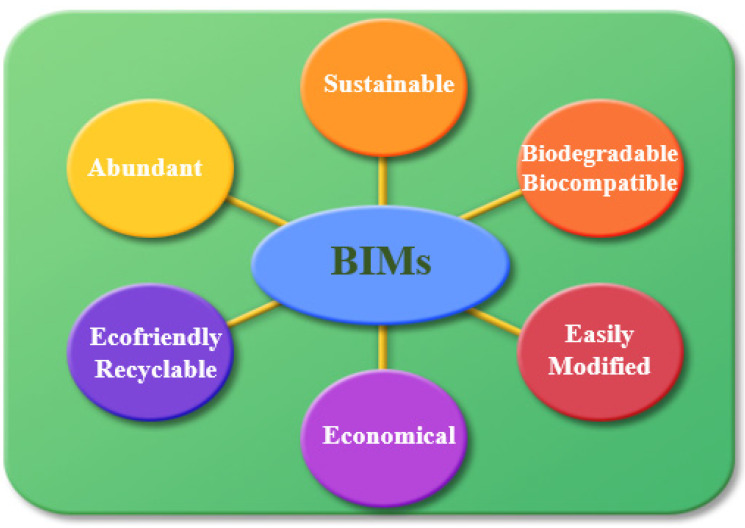
Advantages of biological interfacial materials (BIMs).

**Table 1 micromachines-14-01171-t001:** Electroluminescence performances of OLED devices using biological interfacial layers.

Device Structure	Interfacial Layer	Function	CE (cd A^−1^)	PE (lm W^−1^)	EQE (%)	Ref.
ITO/PEDOT:PSS/DNA–CTMA/NPB/Alq_3_/BCP	DNA–CTMA	Electron blocking layer	8.2	-	-	[50]
ITO/PEDOT:PSS/DNA–CTMA/PFO:MEH-PPV/Cs_2_CO_3_/Al	DNA–CTMA	Hole transport and electron blocking layer	10.0	-	-	[69]
ITO/PEDOT:PSS/poly-TPD:PVK/DNA–CTMA/QDs/TPBi/Alq_3_/Al	DNA–CTMA	Hole transport and electron blocking layer	4.0	-	-	[70]
ITO/PEDOT:PSS/MEH-PPV/DNA/Al	DNA	Electron transport layer	0.15	-	-	[51]
ITO/PEDOT:PSS/Adenine/NPB/CBP:Ir(ppy)_3_/BCP/Alq_3_/LiF/Al	Adenine	Hole transport and electron blocking layer	48	-	-	[62]
ITO/PEDOT:PSS/Thymine/NPB/CBP:Ir(ppy)_3_/BCP/Alq_3_/LiF/Al	Thymine	Hole transport and electron blocking layer	76	-	-	[62]
ITO/PEDOT:PSS/Adenine/CBP:Ir(ppy)_3_/BCP/Alq_3_/LiF/Al	Adenine	Hole/electron blocking layer	51.8	21.2	14.3	[32]
Cellulose/Au/adenine/NPB/CBP:Ir(ppy)_3_/BCP/Alq_3_/LiF/Al	Adenine	Hole injection layer	31.7	-	-	[63]
ITO/PEDOT:ASLS/TAPC/ mCP/mCP:FIrpic/TmPyPb/LiF/Al	Alkyl-Sulfonated Lignin	Hole transportation	37.65	12.84	-	[64]
ITO/PEDOT:GSL/TAPC/mCP/mCP:Flrpic/TmPyPb/LiF/Al	Grafted-Sulfonated Lignin	Hole transportation	26.56	14.67	-	[71]
ITO/Ampicillin-PEDOT:PSS/NPB/TCTA/CBP:Ir(ppy)_3_/TPBi/LiF/Al	Ampicillin (25%)	Hole transportation	120	70	35	[65]
ITO-APC-ITO/Ampicillin-PEDOT:PSS/NPB/TCTA/CBP:Ir(ppy)_2_tmd/TPBi/LiF/Mg:Ag	Ampicillin (25%)	Hole transportation	270	185	68	[68]
ITO/Ampicillin-PEDOT:PSS/TFB/InP-QDs/Ampicillin-ZnO/Al	Ampicillin (25%)	Hole and electron transportation	6.3	6.6	4.7	[72]

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
