# Peer review of "Biological Interfacial Materials for Organic Light-Emitting Diodes"

_micromachines, 2023, doi:10.3390/mi14061171_

Round 1
Reviewer 1 Report
The manuscript reviews the recent development of BIM used for OLED applications. The review provides a valuable guideline for both academic and industrial practitioners to further the use of BIMs for optoelectronic applications. Yet, I have some minor points below for the authors to consider for potential improvement of the manuscript.
- Make sure the font and size of the text is consistent throughout the manuscript, for instance, section 2.1.
- Can the authors add some indicators in Figure 1 of the shown bio-materials for better readability?
- Line 133, it is recommended to add the full name for the acronyms at their appearance.
- Section 4 reviews the state-of-the-art for more materials than introduced in Section 2. Would it be possible to incorporate more materials in Section 2 to have better consistency and completeness?
- In section 4.1, can the authors add more references on the latest progress of Ampicillin as BIM for OLED? Currently only three (3) publications are mentioned and two (2) of them are the authors’ own work.
Some proofreading is required.
Author Response
Thank you for your kind comments and suggestions to improve our work. We have revised the manuscript as per your comments.
- Make sure the font and size of the text is consistent throughout the manuscript, for instance, section 2.1.
Thank you for pointing out the mistake. We have made the font size uniform in the revised manuscript.
- Can the authors add some indicators in Figure 1 of the shown bio-materials for better readability?
Thank you for suggestion. We have added the indicator in the revised manuscript.
- Line 133, it is recommended to add the full name for the acronyms at their appearance.
Thank you for suggestion. We have added the indicator in the revised manuscript.
Line 33 (revised manuscript)
“The major portion of the lignin and ligno-sulfonates is obtained from the paper and is utilized to generate electricity.”
- Section 4 reviews the state-of-the-art for more materials than introduced in Section 2. Would it be possible to incorporate more materials in Section 2 to have better consistency and completeness?
Thank you for your comments. We have incorporated more materials in Section 2 to have better consistency and completeness.
Line 160-170 (revised manuscript)
“2.4. Nucleobases (NBs)
Nucleobases (NBs) of DNA are also known as nucleic acid bases (such as adenine (A), cytosine (C), guanine (G) thymine (T) and uracil (U)). RNA is a single-stranded nucleic acid bio-polymer that serves to covert the DNA base sequence into different proteins. RNA consists of A , C , G , and U bases. The NBs can be obtained from renewable materials and can also be prepared synthetically enabling them a very much economical alternative to DNA and other conventional organic optoelectronic materials. These are basically small molecules that can employed for device fabrication through vacuum vapor deposition method without further purification/modification. Additionally, NBs posses simple molecular structure and much lower molecular weight compared to DNA polymers, which makes their processing very easy.”
- In section 4.1, can the authors add more references on the latest progress of Ampicillin as BIM for OLED? Currently only three (3) publications are mentioned and two (2) of them are the authors’ own work.
Thanks for your suggestion. To the best of our knowledge, no other work has been reported yet regarding ampicillin BIM for OLED device. We have recently pioneered the application of ampicillin BIM for bottom emission and top emission OLED devices.

Reviewer 2 Report
The manuscript entitled “Biological Interfacial Materials for Organic Light-Emitting Diodes”, and author reviewed various BIMs using in OLED, having a discussion associated with different kinds of BIMs and their electrical and physical properties for OLEDs. BIMs are promising materials for green electronics. However, some issues of this manuscript need to be address, I cannot recommend current version of manuscript for publication. The comments are as follows.
1. Author should significantly address the potential of BIM and concisely summarize a perspective for whose future outlook for OLED applications in Abstract.
2. Furthermore, how about status quo of OLED using BIM comparing with conventional OLED? A comparison regarding OLED using BIM and conventional OLED (without BIM) in primary color should be given in their performance including efficiency, lifetime.
3. The function of BIM for OLED should be included in Table 1 for each device. In addition, other than efficiency, lifetime and stability of OLED is also crucial parameter in practical application. Hence, lifetime and stability of OLEDs in Table 1 should be given as well and have some elaboration.
4. Some errors are found in manuscript.
(a) The description of Figure 3 is behind Figure 4. The use of Figure should be in sequential in main text.
(b) Ref. 20 is not found since this citation is not completed.
(c) Some Figures are not clear (e.g. Figure 4a is too small; Resolution of Figure 6 is poor)
Some gammar errors are found in text and minor editing is required
Author Response
Thank you for your kind comments and suggestions to improve our work. We have revised the manuscript as per your comments.
- Author should significantly address the potential of BIM and concisely summarize a perspective for whose future outlook for OLED applications in Abstract.
Thank you for your kind suggestion. We have revised the manuscript with addressing the potential of BIMs.
Line 35-40 (revised manuscript)
Abstract
“Biological materials such as; ampicillin, deoxyribonucleic acid (DNA), nucleobases (NBs) and lignin derivatives have demonstrated significant potential as hole/electron transport layer as well as hole/electron blocking layers for OLED device. Biological materials capable of generating a strong interfacial dipole can be considered as a promising prospect for alternative interlayer materials for OLED applications.”.
- Furthermore, how about status quo of OLED using BIM comparing with conventional OLED? A comparison regarding OLED using BIM and conventional OLED (without BIM) in primary color should be given in their performance including efficiency, lifetime.
Thanks for your comments. We have added a comparison of BIMs and non-BIMs (conventional materials) with the merits/demerits. Regarding color comparison table, most of the authors did not provide the color purity parameters (CIE coordinates). Therefore, we have not added the color purity in the table.
Line 473-494 (revised manuscript)
“To date, various types of interfacial materials have been explored for OLED devices application. These interfacial materials involve metals (Ag, Au, Ca, Al, LiF) and metal oxides (TiO2, ZnO, CsCO3) possessing low WF. Even though these interfacial materials enhanced the efficiency of OLED devices, but they face some serious issues. For instance, metals (Ag, Au, LiF) are prone to air and moisture. However, these are inorganic interfacial materials by nature and their energy levels do not match well with the energy levels of organic layers, which lead to poor charge injection. Also, these are expensive materials and require complicated methods for their processing. Compared to inorganic materials, organic materials are receiving excessive attention as interfacial layers because of multiple advantages (easy solution processability, facile preparation, inexpensive, tunable energy-levels and good stability). Recently, BIMs have shown excellent results and significantly boosted the device efficiency. Some OLED devices containing BIMs (DNA and lignin) have exhibited better stability than that of conventional interfacial materials-based devices. Biological materials possess more structural modification probability compared to other synthetic organic interlayer materials. Additionally, they are abundant, eco-friendly, biodegrable, sustainable and inexpensive. However, a very small number of biological interfacial materials have been applied in OLEDs as compared to other interfacial materials (inorganic and organic). Despite a growing progress of BIMs (synthesis and device application), much more deep studies and investigations are needed to evaluate their complete mechanism of WF modification of electrodes by BIMs”.
- The function of BIM for OLED should be included in Table 1 for each device. In addition, other than efficiency, lifetime and stability of OLED is also crucial parameter in practical application. Hence, lifetime and stability of OLEDs in Table 1 should be given as well and have some elaboration.
Thank you for your kind suggestion. We have added the function of BIMs for OLED in Table 1 in the revised manuscript. Indeed, the lifetime and stability of OLED is also crucial parameter in practical application. However, most of the reports published related to the application of BIMs for OLED, did not provide any data related to lifetime and stability. Therefore, we have not added the comparison of these parameters.
Table 1 (Revised manuscript)
|
Device Structure |
Interfacial layer |
Function |
CE (cd A-1) |
PE (lm W-1) |
EQE (%) |
Ref. |
|
ITO/PEDOT:PSS/DNA-CTMA/NPB/Alq3/BCP |
DNA-CTMA |
Electron blocking layer |
8.2 |
- |
- |
50 |
|
ITO/PEDOT:PSS/DNA-CTMA/PFO:MEH-PPV/Cs2CO3/Al |
DNA-CTMA |
Hole transport and electron blocking layer |
10.0 |
- |
- |
69 |
|
ITO/PEDOT:PSS/poly-TPD:PVK/DNA-CTMA/QDs/TPBi/Alq3/Al |
DNA-CTMA |
Hole transport and electron blocking layer |
4.0 |
- |
- |
70 |
|
ITO/PEDOT:PSS/MEH-PPV/DNA/Al |
DNA |
Electron transport layer |
0.15 |
- |
- |
51 |
|
ITO/PEDOT:PSS/Adenine/NPB/CBP:Ir(ppy)3/BCP/Alq3/LiF/Al |
Adenine |
Hole transport and electron blocking layer |
48 |
- |
- |
62 |
|
ITO/PEDOT:PSS/Thymine/NPB/CBP:Ir(ppy)3/BCP/Alq3/LiF/Al |
Thymine |
Hole transport and electron blocking layer |
76 |
- |
- |
62 |
|
ITO/PEDOT:PSS/Adenine/CBP:Ir(ppy)3/BCP/Alq3/LiF/Al |
Adenine |
Hole/electron blocking layer |
51.8 |
21.2 |
14.3 |
32 |
|
cellulose/Au/adenine/NPB/CBP:Ir(ppy)3/BCP/Alq3/LiF/Al |
Adenine |
Hole injection layer |
31.7 |
- |
- |
63 |
|
ITO/PEDOT:ASLS/TAPC/ mCP/mCP:FIrpic/TmPyPb/LiF/Al |
Alkyl-Sulfonated Lignin |
Hole transportation |
37.65 |
12.84 |
- |
64 |
|
ITO/PEDOT:GSL/TAPC/mCP/mCP:Flrpic/TmPyPb/LiF/Al |
Grafted-Sulfonated Lignin |
Hole transportation |
26.56 |
14.67 |
- |
71 |
|
ITO/Ampicillin-PEDOT:PSS/NPB/TCTA/CBP:Ir(ppy)3/TPBi/LiF/Al |
Ampicillin (25%) |
Hole transportation |
120 |
70 |
35 |
65 |
|
ITO-APC-ITO/Ampicillin-PEDOT:PSS/NPB/TCTA/CBP:Ir(ppy)2tmd/TPBi/LiF/Mg:Ag |
Ampicillin (25%) |
Hole transportation |
270 |
185 |
68 |
68 |
|
ITO/Ampicillin-PEDOT:PSS/TFB/InP-QDs/Ampicillin-ZnO/Al |
Ampicillin (25%) |
Hole and electron transportation |
6.3 |
6.6 |
4.7 |
72 |
- Some errors are found in manuscript.
(a) The description of Figure 3 is behind Figure 4. The use of Figure should be in sequential in main text.
Thanks for pointing out the mistake. We have corrected the sequence of the description of Figures.
Line 214-216 (revised manuscript)
“As far as the use of biological materials in OLED devices is concerned, it mainly includes ampicillin, DNA, nucleobases (NBs) and lignin compounds (Figure 3) till date.”
Line 261-263 (revised manuscript)
“We exploited different ampicillin concentrations and observed that the WF of PEDOT:PSS is reduced by increasing the ampicillin concentration into PEDOT:PSS (Figure 4)”.
(b) Ref. 20 is not found since this citation is not completed.
Line 65-67 (revised manuscript)
“These defects are often produced from either chemical interaction or thermal damage and/or structural imperfection, which results in low device stability. 20-22”
Reference 20
- Suemori, M. Yokoyama and M. Hiramoto, J Appl Phys, 2006, 99.
(c) Some Figures are not clear (e.g. Figure 4a is too small; Resolution of Figure 6 is poor)
Thanks for pointing out the mistake. We have revised the resolutions of the figures in the manuscript.
Figure 4a (Revised Manuscript)
Figure 6 (Revised Manuscript)

Reviewer 3 Report
This review paper focuses on an interesting topic in the field of OLEDs materials. It would be helpful for advancing the design of stable OLEDs and for improving its sustainability by using biological materials. It can be published after minor revisions.
1. It might be better to briefly discuss the currently used interfacial materials in OLEDs, including their structural features, merits and disadvantages. This can highlight the significance of developing BIM.
2. Page 3, lines 11-121. The characters are different. Please revise.
Author Response
Thank you for your kind suggestion/comments to improve the quality of our work. We have revised the manuscript as per your suggestions.
- It might be better to briefly discuss the currently used interfacial materials in OLEDs, including their structural features, merits and disadvantages. This can highlight the significance of developing BIM.
Thanks for your suggestion. We have briefly added the description of the currently used interfacial layers.
Line 473-494 (revised manuscript)
To date, various types of interfacial materials have been explored for OLED devices application. These interfacial materials involve metals (Ag, Au, Ca, Al, LiF) and metal oxides (TiO2, ZnO, CsCO3) possessing low WF. Even though these interfacial materials enhanced the efficiency of OLED devices, but they face some serious issues. For instance, metals (Ag, Au, LiF) are prone to air and moisture. However, these are inorganic interfacial materials by nature and their energy levels do not match well with the energy levels of organic layers, which lead to poor charge injection. Also, these are expensive materials and require complicated methods for their processing. Compared to inorganic materials, organic materials are receiving excessive attention as interfacial layers because of multiple advantages (easy solution processability, facile preparation, inexpensive, tunable energy-levels and good stability). Recently, BIMs have shown excellent results and significantly boosted the device efficiency. Some OLED devices containing BIMs (DNA and lignin) have exhibited better stability than that of conventional interfacial materials-based devices. Biological materials possess more structural modification probability compared to other synthetic organic interlayer materials. Additionally, they are abundant, eco-friendly, biodegrable, sustainable and inexpensive. However, a very small number of biological interfacial materials have been applied in OLEDs as compared to other interfacial materials (inorganic and organic). Despite a growing progress of BIMs (synthesis and device application), much more deep studies and investigations are needed to evaluate their complete mechanism of WF modification of electrodes by BIMs.
- Page 3, lines 11-121. The characters are different. Please revise.
Thanks for pointing out the mistake. We have revised the characters in the manuscript.

Reviewer 4 Report
The authors reviewed biological interfacial materials (BIMs) and their evolution in the organic light emitting diode (OLED) devices by highlighting the electrical and physical properties. The review is timely good because many researchers can follow to devlop such BIMs for OLED displays in near future, so I recommend to accept this paper by addressing some minor comments.
1. Author should discuss and cite state of the art works such as Adv. Mater. 2022, 34, 2202866 ; Sci Rep 4, 7105 (2014).
2. The font is not uniform thought out the manuscript. Revise it clearly.
3. It would be great if authors include the picture/table indicating the key advantages.
4. The authors should emphasize about the degradability issues relevant to OLEDs.
5. Only limited BIMs are discussed, I recommend to add others as well.
6. In case of DNA based BIMs, Is it possible to tune the color with control of voltage by taking help of some sort of Ruthinium (Ru) complexes. There were some reports on it. Eloborate it more.
Author Response
Thank you for your kind comments and suggestions to improve our work. We have revised the manuscript as per your comments.
- Author should discuss and cite state of the art works such as Mater. 2022, 34, 2202866 ; Sci Rep 4, 7105 (2014).
We have already discussed and cited these state of the art works in the manuscript. This (Adv. Mater. 2022, 34, 2202866) is our own work and we have discussed this and other work (Sci Rep, 2014, 4, 7105) in the manuscript in detail as follows.
Line 264-275 (In manuscript) (Adv. Mater. 2022, 34, 2202866)
“Inspired from these results, the same group have very recently optimized the annealing conditions to reduce the variations in the efficiency and stability of ampicillin based top-emission OLEDs (TEOLEDs).68 By changing the annealing conditions, the formation of Amp-microstructure (Amp-MSs) of different sizes and shapes (α-/β-phase) led to different energy states. Amp-MSs excited the light-scattering/out-coupling of the device, which resulted in the reduction of the waveguide modes. 68 TEOLED device exhibited a record-high EQE (maximum: 68.7% and average: 63.4% at spectroradiometer; maximum: 44.8% and average: 42.6% at integrating sphere) with a wider color gamut (118%) due to the red-shifted J-aggregated emission. An exceptional increase in the efficiency was assigned to the improved charge balance and out-coupling, formation of interfacial dipole, photoluminescence through radiative energy transfer (RET) and EL by J-/H-aggregated excitons (Figure 4e-4f)”
Line 364-378 (In manuscript) Sci Rep 4, 7105 (2014)
“Thymine and adenine bases are among those four nitrogenous bases that constitute a large DNA polymer. Thymine belongs to the pyrimidine class and contains one heterocyclic-ring, whereas adenine is a member of purine class possessing two fused rings. These NBs do not involve surfactant treatment or wet processing to make thin films. Green phosphorescent OLED using thymine as an EBL/HTL, obtained 76.0 cd A-1 CE, which demonstrated 200% improvement in efficiency compared to the device without NB (37.0 cd A-1) (Figure 5c-5d).62 While, adenine based device also surpassed the reference device with a maximum CE of 48.0 cd A-1. A low efficiency roll-off was also observed at higher voltages, resulting in enhanced efficiency (Figure 5c-5d). A similar roughness was found in both bases (1.76 and 1.83 nm).62 AFM scan of adenine showed a more uniform distribution of crystallites with respect to periodicity and height-distribution. Thymine crystallites exhibit columnar-like structures, while adenine crystallites possess lower height dispersion. A better performance of thymine-based device is ascribed to a coalition of energy levels and structured surface morphology which outcomes in efficient hole transportation to EML”.
- The font is not uniform thought out the manuscript. Revise it clearly.
Thank you for pointing out the mistake. We have made the font size uniform in the revised manuscript.
- It would be great if authors include the picture/table indicating the key advantages.
Thank you for your suggestion. We have added a picture elaborating the key advantages of BIMs.
- The authors should emphasize about the degradability issues relevant to OLEDs.
Thank you for your kind suggestion. We have highlighted the degradability issues related to OLED in the revised manuscript.
Line 521-533 (revised manuscript)
An enhanced performance stability of OLEDs can be realized by overcoming the challenge of device degradation evoked by environmental, operational, and fabrication-process parameters. Firstly, environmental degradation of device generally takes place due to the exposure of device from oxygen and moisture. Encapsulation with a suitable sealant has been employed to solve this problem. Secondly, operational degradation is also an important problem caused by the WF mismatch between the electrodes and connected organic layers, which leads to the production of pin–holes, and heat is produced internally through triplet–triplet annihilation (TTA) or triplet–polaron annihilation (TPA). Lastly, from the processability point of view, several factors (substrate cleaning, solvent, and annealing process) also play a key role to increase the stability of OLEDs. BIMs based devices have exhibited excellent stability compared to the reference (conventional device). However, more work needs to be done to investigate their long-term stability potential for commercial applications.
- Only limited BIMs are discussed, I recommend to add others as well.
Thank you for your suggestion. Indeed, several BIMs have been used in organic optoelectronic devices recently. However, most of those have been employed in organic and perovskite photovoltaic devices. Herein, we have highlighted those BIMs only which have been used in OLED applications (those are very less as we have reported in this review).
- In case of DNA based BIMs, Is it possible to tune the color with control of voltage by taking help of some sort of Ruthinium (Ru) complexes. There were some reports on it. Eloborate it more.
Thanks for your suggestion. Yes, it is it possible to tune the color with control of voltage by taking help of some sort of Ruthinium (Ru) complexes. We have elaborated it in the manuscript also.
Line 335-341 (In manuscript)
“Later on, DNA was also integrated with polyaniline to form a metal complex [DNA/Polyaniline/Ru(bpy)32+] to be employed as HTL for OLED device.76 OLED with a configuration ITO/DNA/Polyaniline/Ru(bpy)32+/Alq3/Al showed green light with a high VON of 5 V. By accelerating voltage, emission was altered from green to yellow (14 V) to orange (16 V) and finally to red (18 V). An appreciable role of green emission was due to Alq3 and an important role of red emission originated from Ru(bpy)32+.”

Round 2
Reviewer 2 Report
Reviewer's comments have been addressed, the revised manuscript can be accepted for publication.